# Current and Future Landscape of Perioperative Treatment for Muscle-Invasive Bladder Cancer

**DOI:** 10.3390/cancers15030566

**Published:** 2023-01-17

**Authors:** Jorge Esteban-Villarrubia, Javier Torres-Jiménez, Carolina Bueno-Bravo, Rebeca García-Mondaray, José Daniel Subiela, Pablo Gajate

**Affiliations:** 1Medical Oncology Department, 12 de Octubre University Hospital, 28041 Madrid, Spain; 2Medical Oncology Department, Hospital Clínico San Carlos, 28040 Madrid, Spain; 3Urology Department, Infanta Sofía University Hospital, 28702 Madrid, Spain; 4Urology Department, San Pedro Hospital, 26006 Logroño, Spain; 5Urology Department, Ramon y Cajal University Hospital, 28034 Madrid, Spain; 6Medical Oncology Department, Ramon y Cajal University Hospital, Instituto Ramón y Cajal de Investigación Sanitaria (IRYCIS), 28034 Madrid, Spain

**Keywords:** muscle-invasive bladder cancer, neoadjuvant, adjuvant, chemotherapy, immunotherapy, targeted agents

## Abstract

**Simple Summary:**

The risk of recurrence of patients with localized muscle-invasive bladder carcinoma (MIBC) is still high. The outcomes of surgery and perioperative therapy are limited, and several patients are not candidates for neoadjuvant chemotherapy and have no further alternatives available. In recent years, many drugs have been evaluated in the metastatic setting. This review summarizes the evidence of perioperative treatment with these new drugs for MIBC, emphasizing immunotherapy and targeted agents.

**Abstract:**

Cisplatin-based neoadjuvant chemotherapy followed by radical cystectomy is the current standard of care for muscle-invasive bladder cancer (MIBC). However, less than half of patients are candidates for this treatment, and 50% will develop metastatic disease. Adjuvant chemotherapy could be offered if neoadjuvant treatment has not been administered for suitable patients. It is important to reduce the risk of systemic recurrence and improve the prognosis of localized MIBC. Systemic therapy for metastatic urothelial carcinoma has evolved in recent years. Immune checkpoint inhibitors and targeted agents, such as antibody-drug conjugates or FGFR inhibitors, are new therapeutic alternatives and have shown their benefit in advanced disease. Currently, several clinical trials are investigating the role of these drugs, as monotherapy and in combination with chemotherapy, in the neoadjuvant and adjuvant settings with promising outcomes. In addition, the development of predictive biomarkers could predict responses to neoadjuvant therapies.

## 1. Introduction

Bladder cancer (BC) is the 10th most commonly diagnosed cancer worldwide, with approximately 573,000 new cases and 213,000 deaths each year [1]. Muscle-invasive bladder cancer (MIBC) represents around 20% of newly diagnosed cases of BC [2]. Radical cystectomy (RC) with pelvic lymph node dissection is the standard treatment in the MIBC setting. Cisplatin-based neoadjuvant chemotherapy (NC) has demonstrated a benefit in overall survival (OS) [3]. However, only 25–50% of patients are fit for this therapy, and many of them continue to undergo RC up front [4]. Despite surgery and systemic therapy, approximately 50% of patients will relapse within 2 years [5]. Therefore, an improvement in the management of localized disease is needed to reduce systemic recurrence in these patients.

In recent years, the treatment of metastatic urothelial carcinoma (UC) has changed rapidly, with several drugs approved in this setting. Immune checkpoint inhibitors (ICI) targeting PD-1 and PD-L1 have been established as standard therapies in advanced UC. Currently, ICI are approved for maintenance therapy after a response or stable disease to first-line platinum-based chemotherapy as first-line therapy in cisplatin-ineligible patients with a PD-L1 positive tumor and for platinum-ineligible and refractory patients [3]. Due to their favorable safety profile in comparison with cytotoxic drugs and their possibility to combine with chemotherapy, ICI are being investigated in the perioperative setting in cisplatin-ineligible and eligible patients. In addition, antibody-drug conjugates (ADC), such as enfortumab vedotin or sacituzumab govitecan, and targeted agents, such as erdafitinib, have shown promising results that are changing the landscape of the management of UC with potential impacts in the treatment algorithm [6,7,8]. 

This article reviews the current evidence of perioperative management of MIBC, presents the data of ICI and new therapies and discusses potential predictive biomarkers for the different treatments.

## 2. Neoadjuvant Treatment

### 2.1. Neoadjuvant Chemotherapy (NAC)

Currently cisplatin-based chemotherapy before surgery is the standard of care in the management of MIBC. NAC should be recommended for all patients with T2 to T4 or N1 MIBC who are eligible for cisplatin [3,9]. Two randomized clinical trials and meta-analysis have shown an OS benefit with this strategy compared with up-front RC. In addition, alternative chemotherapy regimens have been evaluated in the neoadjuvant setting (Table 1).

Two phase 3 randomized clinical trials have shown a longer OS with cisplatin-based chemotherapy in the neoadjuvant setting. The SWOG-8710 trial included 317 patients with T2-T4aN0 MIBC. They were randomized to three cycles of neoadjuvant methotrexate, vinblastine, doxorubicin, and cisplatin (MVAC) followed by surgery compared to RC alone [10]. A benefit in OS was demonstrated, with a median OS (mOS) of 77 months in the neoadjuvant group vs. 46 months in the surgery alone group (*p* < 0.06). In addition, an increase in the pathologic complete response, defined as pT0N0, was observed with neoadjuvant chemotherapy (38% vs. 15%; *p* < 0.001). In this study, pCR was described as a surrogate measure of OS due to the excellent outcomes in these patients; 85% of them were free of disease at 5 years. The BA06 30,894 trial evaluated three cycles of cisplatin, methotrexate and vinblastine in the neoadjuvant setting compared to local therapy alone (RC or radiotherapy) [11]. More than 900 patients were included. In the first analysis, an improvement in OS was not observed. However, with a longer follow-up a statistically significant 16% reduction in the risk of death was seen with chemotherapy before surgery (hazard ratio (HR), 0.84; 95% CI, 0.72 to 0.99; *p* = 0.037), with a longer 10-year OS (30% vs. 36%).

A modified MVAC regimen (dose-dense MVAC (dd-MVAC)) has also been tested in two phase 2 clinical trials. Choueiri et al. evaluated four cycles of dd-MVAC as neoadjuvant therapy in 39 patients with MIBC in a single-arm phase 2 trial [12]. The primary endpoint was pathologic response (PaR) defined by pathologic downstaging to ≤pT1N0. The PaR was 49% (80% CI, 38 to 61), with a 26% of pT0N0. Grade 3 or higher toxicity was observed in 10% of patients, with no neutropenic fevers or treatment-related death. Plimack et al. administered three cycles of neoadjuvant accelerated MVAC in a phase 2 trial with 44 patients [13]. This study reported 38% of pCR and 52% of downstaging to non-muscle invasive disease. In addition, most of the patients (82%) experienced only grade 1–2 treatment-related toxicities. There were no treatment-related deaths. Currently, dd-MVAC has replaced classic MVAC because of its better tolerance, shorter duration and higher pCR rate.

Cisplatin plus gemcitabine (CG) is the standard first-line treatment in metastatic UC due to a phase 3 trial that showed a similar progression-free survival (PFS) and OS compared to MVAC with a better safety profile [14]. Despite the lack of randomized trials with preoperative CG, these data have been extrapolated to the neoadjuvant setting, and CG is commonly used prior to surgery in MIBC patients. Some retrospective studies have described similar outcomes with this regimen as neoadjuvant therapy [15,16,17].

Various meta-analyses have demonstrated a benefit in OS with cisplatin-based chemotherapy vs. up-front surgery in patients with MIBC. GETUG-AFU V05 VESPER trial is a phase III randomized study that compares dd-MVAC vs. CG in patients with MIBC as perioperative therapy [18,19]. GETUG-AFU V05 VESPER trial is a phase III randomized study that compares dd-MVAC vs. CG in patients with MIBC as perioperative therapy [20]. A total of 88% of patients received chemotherapy in the neoadjuvant setting. The primary endpoint was PFS at 3 years. In the neoadjuvant group, PFS at 3 years was significantly longer in the dd-MVAC arm (66% vs. 56%, HR 5 0.70 [95% CI, 0.51 to 0.96], *p* = 0.025). In addition, dd-MVAC reached a higher local control rate (pCR, tumor downstaging or organ confined) 63% vs. 50% (*p* = 0.021).

**Table 1 cancers-15-00566-t001:** Summary of trials for neoadjuvant chemotherapy in MIBC.

	SWOG-8710 [10]	BA06 30,894 [11]	Choueiri et al. [12]	Plimack et al. [13]	Dash et al. [15]	Iyer et al. [16]	GETUG-AFU V05 VESPER [20]
*n*	317	976	39	40	42	154	437
Phase	3 Ra	3 Ra	2 SA	2 SA	Re	Re	3 Ra
Treatment	MVAC vs. surgery	CMV vs. local therapy	dd-MVAC	aa-MVAC	CG	CG	dd-MVAC vs. CG
pCR (pT0N0)	38%	NA	26%	38%	26%	21%	42% vs. 36%
downstaging to non-MIBC (<pT2)	44%	NA	49%	53%	36%	46%	63% vs. 50%
OS	mOS: 77 months vs. 46 months (*p* = 0.06)	10 y OS rate: 30% vs. 36% (HR = 0.84; (95% CI, 0.72 to 0.99; *p* = 0.037)	2 y OS rate: 79%	2 y OS rate: 83%	2 y OS rate: 73%	2 y OS rate: 72%	NR vs. NR (HR = 0.66 (95% CI, 0.47 to 0.92)

Abbreviations: Ra: randomized, SA: single-arm, Re: retrospective, MVAC: methotrexate, vinblastine, doxorubicin and cisplatin, dd-MVAC: dose-dense methotrexate, vinblastine, doxorubicin and cisplatin, aa-MVAC: accelerated methotrexate, vinblastine, doxorubicin and cisplatin, CG: cisplatin and gemcitabine, pCR: pathological complete response, NA: not available, OS: overall survival, mOS: median overall survival, y: years, HR: hazard ratio.

Despite these data, neoadjuvant treatment remains underutilized, many patients are not suitable to cisplatin-based neoadjuvant chemotherapy and the number of patients that will recur with this approach is still very high.

### 2.2. Single Inmmunotherapy Agents

Several trials to date have addressed the benefit of immunotherapy agents in the perioperative treatment of MIBC (Table 2). The PURE-01 trial was an open-label, single-arm, phase-2 trial that included 50 patients with a predominant (at least 50%) UC histology and clinical T3b or less of an N0 stage tumor. Patients were included regardless of their cisplatin eligibility, and 92% of included patients were cisplatin-eligible. Patients were administered three cycles of pembrolizumab before surgery. The primary endpoint of the trial was a pCR rate of 25%. This endpoint was achieved as pCR in the study was 42%. A total of 52% of patients achieved the down-staging of the primary tumor [21]. Updated survival analyses of this trial evidenced a 1 and 2 year event-free survival (EFS) of 84.5% and 71.7%, respectively [22]. Only 6% of patients had grade 3 immune-related adverse events (irAEs), with only one patient that required treatment discontinuation. The ABACUS trial included 95 patients with MIBC (T2-T4aN0M0) who were ineligible or refused neoadjuvant cisplatin treatment. These patients received two cycles of Atezolizumab before RC. Of these 95 patients, 85 underwent surgery. The pCR rates were 31% in all patients and 37% of PD-L1-positive patients. In an updated report of the trial results, 2-year disease-free survival (DFS) was 68%. In patients who achieved pCR, the 2-year DFS improved to 85%. Higher T stage at baseline and at cystectomy and node-positive correlated with poor DFS. The 2-year OS rate was 77% [23]. The results of the cisplatin-ineligible arm of the AURA trial (Oncodistinct-004) have recently been presented in ASCO 2022 meeting. This arm included patients with cT2-T4aN0-2M0 UC that were randomized to avelumab monotherapy or paclitaxel (P) + gemcitabine (G) + avelumab. The pCR was achieved in 16% of patients in the P + G+ avelumab arm, compared to 36% in the avelumab monotherapy arm. Downstaging showed a similar trend between arms. Ongoing studies with single immunotherapy approaches will help to clarify the role of neoadjuvant single immunotherapy agents in this population.

Regarding other histologies, after an amendment in March 2018, the PURE-01 trial allowed the inclusion of predominant variant histologies (VH). An update of the results of the trial, including these patients (*n* = 19), has recently been published. A substantially lower pCR rate was found (16%) in this subgroup. Responses were achieved in squamous cell carcinoma and lymphoepithelioma-like variant patients, while no pathological responses were noted in the other variants. However, authors suggest that responses were related to tumor biomarkers as responding VH patients were enriched in higher TMB and CPS patients [24]. In the ABACUS trial, only patients with predominant urothelial carcinoma were included, so data in this subgroup of patients is scarce. The ABACUS-2 trial is currently ongoing and recruiting patients in two different arms: One for urothelial carcinoma of the upper tract and the other for rarer histological subtypes (NCT04624399). In addition, the EV-303/KEYNOTE-905 trial will compare pembrolizumab monotherapy vs. pembrolizumab + enfortumab vedotin (EV) vs. up-front surgery (NCT03924895).

### 2.3. Combination of Inmmunotherapy Agents

Anti-PD(L)1 agents have been usually combined with anti-CTLA4 drugs to potentiate each other. In UC, this combination in the neoadjuvant scenario have been investigated in several trials (Table 2). In the NABUCCO trial, the combination of nivolumab and ipilimumab was addressed. A total of 24 patients with stage III UC were treated in this trial with a defined sequence of ipilimumab (3 mg/kg in days 1 and 22) and nivolumab (3 mg/kg in days 22 and 43) followed by RC in the cohort 1. As a feasibility trial, the main endpoint was the possibility to resect within 12 weeks of starting the treatment, showing a 96% resection rate. A total of 46% of the patients achieved a pCR and 58% showed no invasive residual tumor. Grade 3–4 immune-related adverse events (irAEs) were present in 55% of patients [25]. The aim of NABUCCO cohort 2 was to find an optimal dose of preoperative nivolumab and ipilimumab. Patients in cohort 2a received two cycles of nivolumab at 1 mg/kg with ipilimumab 3 mg/kg followed by a third cycle of nivolumab at 3 mg/kg. In the cohort 2b, the treatment was nivolumab at 3 mg/kg in combination with ipilimumab 1 mg/kg and an extra cycle of nivolumab at 3 mg/kg. The pCR in cohort 2a was higher in comparison with cohort 2b (43% vs. 7%) [26]. In addition, a higher G3-4 irAEs were observed in cohort 2a patients (33% vs. 20%). Another trial also investigated different nivolumab and ipilimumab schedules in cisplatin-ineligible patients. This study was designed to compare three cohorts of patients. In the first cohort, nivolumab was administered at 3 mg/kg every two weeks for five cycles. In the second cohort, nivolumab at 1 mg/kg in combination with ipilimumab 3 mg/kg was administered at weeks 0 and 6 of treatment, while nivolumab 3 mg/kg was administered in weeks 3 and 9. In the third cohort, three cycles of nivolumab 1 mg/kg with ipilimumab 3 mg/kg were planned. The primary endpoint of the trial was eligibility for planned RC within 60 days, without delay from treatment-related adverse events. Cohort 1 met the primary endpoint with 14 patients of 15 undergoing surgery, while in the cohort 2 only 8 patients of 15 underwent surgery. Accrual for the third cohort stopped due to the cohort 2 results. The pCR in these cohorts were 13% and 7%, respectively [27].

The DUTRENEO trial included a population of 61 cisplatin-eligible patients with cT2a-T4aN0-1 tumors. The patients included in this trial had a tumor pro-inflammatory interferon-𝛾 (INF-𝛾) immune signature assessed and divided into “hot” and “cold” tumors. Patients with “hot” tumors were randomized to standard cisplatin-based chemotherapy or to three cycles of immunotherapy, while “cold” patients received chemotherapy. This approach failed to select the patients more responsive to immunotherapy. Patients with “hot” tumors showed a similar pCR with immunotherapy or chemotherapy (34.8% vs. 36.4%, respectively). In addition, pCR in patients with “cold” tumors treated with chemotherapy was 68.8% [28]. A combination study with durvalumab and tremelimumab in cisplatin-ineligible high-risk patients (high-risk features were defined by T3-T4 tumors, hydronephrosis, pathologic variant urothelial carcinoma, vascular and lymphatic invasion, and/or high-grade upper tract disease) showed an overall pCR of 37.5% and downstaging of 58%. In VH, pCR was 57%, although there were only seven patients with VH included [29].

**Table 2 cancers-15-00566-t002:** Summary of completed, active, or currently recruiting studies with immunotherapy agents without chemotherapy.

Study	Phase	Treatment	Patients Included	pCR	Survival
PURE-01 ^1^ [21]	2	PEM	80	39%	-
PURE-01 (VH)	2	PEM	19	16%	-
PANDORE [30]	2	PEM	34	29.4%	-
ABACUS	2	AZ	95	31%	1 y RFS: 79%
AURA (Cohort 2)	2	A; PG + A	56	36% (A) vs. 18% (PG + A)	-
NABUCCO (cohort 1)	2	N + IPI	24	46%	-
NABUCCO (cohort 2)	2	2a: N (1) + IPI (3) 2b: N (3) + IPI (1)	15 15	43% 7%	
MDACC	2	DU + TRE	28	37.5%	1 y RFS: 82.8% 1 y OS: 88%
MDACC (VH)	2	DU + TRE	7	57%	-
DUTRENEO	2	DU +TRE, PG	61	“Hot” arm: 34.8% (DU + TRE) vs. 36.4 (PG) “Cold” arm: 68.8% (PG)	-
CA209-9DJ	2	Cohort 1: N (3)/Cohort 2: N(1) + I(3)	30	Cohort 1: 13% Cohort 2: 7%	12 m RFS C1: 77% 12 m RFS C2: 68%.
ABACUS-2 (VH)	2	Atezolizumab	-	-	-
NCT02845323	2	N ± Urelumab	-	-	-
NCT03532451	1b	N ± Lirilimab	-	-	-
PIVOT-IO 009	3	N ± Bempegaldesleukin	-	-	-
OPTIMUS	2	Retifanlimab, Epacadostat, INCAGN02385, INCAGN02390	-	-	-
BLASST-2	2	PEM + Oleclumab	-	-	-

1: These results correspond to updates results published in 2020. Abbreviations: A: avelumab, AZ: atezolizumab, DU: durvalumab, I: ipilimumab, N: nivolumab, pCR: pathological complete response, PG: cisplatin-gemcitabine, PEM: pembrolizumab, OS: overall survival, RFS: recurrence-free survival, TRE: tremelimumab.

Other strategies include the combination of immunotherapy with novel immune checkpoint inhibitors (ICIs) as antiCD137 (NCT02845323), anti-killer cell immunoglobulin-like receptor (KIR) (NCT03532451), immunostimulatory interleukin-2 (IL-2) cytokine prodrug (PIVOT IO 009, NCT04209114), Indoleamine 2,3-dioxygenase (IDO) inhibitors, anti-lymphocyte-activation gene 3 (LAG3) antibodies, anti-T-cell immunoglobulin and mucin-domain containing-3 (TIM-3) drugs (OPTIMUS, NCT04586244) or antiCD73 (BLASST-2 trial, NCT03773666).

### 2.4. Combination of Inmmunotherapy and Chemotherapy

As the standard of care in the perioperative treatment of MIBC is nowadays cisplatin-based chemotherapy, several trials have addressed if the addition of immunotherapy could improve the results of chemotherapy alone (Table 3). The HCRN GU 14-188 was a phase 1b/2 trial that included patients with T2-T4aN0M0 BC. The study included two cohorts of patients, one for patients eligible for treatment with cisplatin (43 patients) and another for ineligible patients (37 patients). In the cisplatin-eligible cohort, patients received four cycles of cisplatin + gemcitabine overlapped with five cycles of pembrolizumab. In this cohort, the pCR rate was 44.4% [31]. In the cisplatin-ineligible cohort, patients received three cycles of weekly gemcitabine and overlapped with five cycles of pembrolizumab. The pCR in this population was 45.2% [32]. The LCCC1520 trial included 39 cisplatin-eligible patients with a pCR of 36%. In this study, the first six patients received lead-in pembrolizumab 200 mg once 2 weeks, followed by pembrolizumab 200 mg once on day 1, in combination with cisplatin and gemcitabine for four cycles. This regimen was discontinued due to toxicity and subsequent patients received cisplatin 35 mg/m2 on days 1 and 8. This could explain the differences between pCR rates with the previous study [33].

Another study with the introduction of Atezolizumab before four cycles of full dose CG and a consolidation dose of Atezolizumab showed an encouraging 41% of pCR. After a median follow up of 16.5 months, no relapse in patients who achieved a <ypT2N0 was seen. It is important to note that all patients underwent RC [34]. The BLASST-1 was a phase 2 single-arm trial that investigated neoadjuvant CG for four cycles with nivolumab. The cT2a-T4 tumors and node positive patients were allowed. The primary objective of the study was ≤ypT1N0 pathological response, observed in 65.8% of patients. The pCR rate was 34% [35]. At a median follow-up of 15.8 months, 12-month RFS rate was 85.4% and PFS was 83% [36]. Durvalumab has also been tested in combination with cisplatin and gemcitabine in the SAKK 06/17 trial that included node-positive patients, with a pCR of 34% and an EFS at 2 years of 76.1% that was the primary endpoint of the trial [37].

The other regimen of neoadjuvant chemotherapy, which is being tested in combination with immunotherapy, is MVAC. Cohort 1 of the AURA trial assessed the combination of avelumab with CG and dd-MVAC [38]. The pCR was 32% and 43% and the downstaging to ≤ypT1N0 was 57% and 64% with CG and dd-MVAC, respectively, in combination with avelumab. In addition, dd-MVAC will be tested in combination with durvalumab in a phase 2 trial that is currently recruiting (NEMIO trial) [39]. Other immunotherapy agents that are being investigated in combination with MVAC are nivolumab and pembrolizumab (RETAIN-2; NCT04506554 and NCT04383743, respectively).

Other chemotherapies are being evaluated in combination with ICI. The SWOG GAP trial is a phase 2 trial that will include 196 cisplatin-ineligible patients and will compare the combination of carboplatin with gemcitabine and avelumab to observation before surgery (NCT04871529). The combination of tislelizumab, an anti-PD-1 antibody, with Nab-paclitaxel will be addressed in a phase 2 trial of MIBC not extended to regional lymph nodes. In this trial, the primary objective will be the CR rate, but patients will be able to undergo transurethral resection of bladder tumor (TURBT) instead of RC. The results of the cohort in which TURBT was performed have been published. A total of 22 patients were included in this cohort; a 77% pT0, 4.5% pTa, 13.63% pT1 and 4.5% pTis was observed. The recurrence-free survival rate of these patients after one year was 82% [40]. However, patients treated with TURBT continued to receive medication after the procedure, and this may confound the trial outcome and a longer follow-up is needed.

Definitive results from ongoing phase 3 studies will establish the role of chemo-immunotherapy in the perioperative setting. The KEYNOTE 866 is a phase 3 randomized trial of neoadjuvant chemotherapy with perioperative pembrolizumab or placebo in cisplatin-eligible patients. Patients will receive four cycles of cisplatin + gemcitabine in combination with pembrolizumab or placebo, followed by RC plus pelvic lymph node dissection (PLND), and postoperative 13 cycles of adjuvant pembrolizumab or placebo [41]. The NIAGARA trial is an ongoing phase 3 trial with a similar design to KEYNOTE 866, which will address the efficacy of durvalumab in combination with CG in the neoadjuvant setting and durvalumab in the adjuvant [42]. The ENERGIZE trial is another phase 3 trial that will compare the perioperative combination of chemotherapy with nivolumab with or without linrodostat mesylate, an IDO inhibitor [43].

### 2.5. Combination of Inmmunotherapy and Other Systemic Therapies

#### 2.5.1. Combination of Immunotherapy with Antibody–Drug Conjugates

Antibody–drug conjugates (ADCs) are composed by a monoclonal antibody that targets a tumor-associated antigen linked to a cytotoxic payload. Enfortumab Vedotin (EV) is an ADC that targets the membrane Nectin-4, a member of a family of related immunoglobulin-like adhesion molecules, and this receptor is overexpressed in approximately 60% of samples of BC. The antibody is conjugated to monomethyl auristatin E, a microtubule-disrupting agent [44]. As this compound has shown promising results in the metastatic setting [7], EV is currently being investigated in the perioperative setting in combination with pembrolizumab in cis-eligible (EV-304/KEYNOTE-B15 trial, NCT04700124) and ineligible (EV-303/KEYNOTE-905 trial, NCT03924895) patients and in combination with durvalumab and tremelimumab in cis-ineligible patients (VOLGA trial, NCT04960709). Sacituzumab govitecan (SG) is another ADC target to tumor-associated calcium signal transducer 2 (TROP-2)-expressing cancer cells, which is conjugated with SN-38, an anti-topoisomerase 1, as the cytotoxic compound. This ADC has shown positive results in advanced UC [8] and randomized phase III trials are currently recruiting in these patients (TROPICS-04, NCT04527991). A phase 2 trial for cis-ineligible patients in the neoadjuvant setting combining SG with pembrolizumab is currently recruiting [45].

#### 2.5.2. Other Targeted Agents

Fibroblast growth factor receptor (FGFR) is a family of tyrosine kinase receptors constituted by four members: FGFR1–FGFR4. Multiple ligands of the FGF family can bind this receptor, activating downstream transduction intracellular signaling pathways. Enrichment in FGFR3 expression has been shown in luminal subtype tumors and this subtype may represent up to 20% of pT2 tumors [46]. The FGFR inhibition in advanced UC have shown promising results [6], and it supports the development of clinical trials in the neoadjuvant (NCT04228042) and adjuvant settings (PROOF-302 trial), focused in patients not eligible for cisplatin [47]. Furthermore, data from early phase clinical trials suggest that ICIs may increase responsiveness to FGFR inhibitors, and this may represent a rationale to develop trials that combine FGFR inhibitors with ICIs in the neoadjuvant setting of UC in the future [48].

Poly ADP-ribose polymerase (PARP) has a key role in DNA repair and can act as a transcription modulator of genes involved in chromatin remodeling and gene transcription. The decreased expression of genes associated with homologous recombination repair (HRR) pathway results in sensitivity to treatment with PARP inhibitors, causing synthetic lethality [49]. NEODURVARIB was a phase 2 trial that assessed the value of the addition of Olaparib to durvalumab for two cycles in cT2–T4aN0 cisplatin-ineligible patients. The primary endpoint of the study was pCR that was 44.5%. Combination was well tolerated, with adverse events grade 3–4 detected in only 8.3% of cases [50].

An overexpression of hypoxia inducible factor (HIF-1) and vascular endothelial growth factor (VEGF) has been associated with poor prognosis and metastatic spread in urothelial carcinoma. Vessel density is associated with vascular invasion, recurrence and shorter survival in invasive UC [51]. Cabozantinib, a VEGFR2 and MET inhibitor, has demonstrated immunomodulatory results, a decrease in the number of myeloid-derived suppressor cells and regulatory T cells and an increase in PD-1 expression in regulatory T-cells. This may lead to a less immunosuppressive stroma and may be able to potentiate anti-PD1 immunotherapy effects [52]. To explore this hypothesis, the phase 2 ABATE trial will include cT2–T4aN0/xM0 cisplatin-ineligible patients. The primary objective of the study is to detect more than 20% of <pT2 response [53]. Methionine aminopeptidase 2 inhibition has been shown to inhibit the proliferation of human microvascular endothelial cells and tumor cells [54], and it has been hypothesized that this effect could be synergistic with PD-1 inhibition. The ANTICIPATE trial is a phase I/II trial that will test the combination of APL-1202 with Tislelizumab as a neoadjuvant therapy in MIBC [55].

CD38 is a membrane receptor expressed in tumor-associated macrophages and inhibits lymphocyte T CD8 function via adenosine receptor signaling. CD38 overexpression has been associated with resistance to ICIs [56], and CD38 blockade has been shown to suppress bladder cancer growth in vivo [57]. Daratumumab, an anti CD38 antibody, is being investigated in the neoadjuvant setting in MIBC (NCT03473730).

Other studies are aiming to find an adequate partner for immunotherapy as the PECULIAR trial or NCT03978624. Both will combine pembrolizumab with epigenetic modifiers, such as Epacadostat or Entinostat. The other approach is the combination of ICIs with a replication-competent oncolytic adenovirus in cisplatin-ineligible patients with MIBC (NCT04610671).

## 3. Adjuvant Treatment

### 3.1. Chemotherapy

Several studies have investigated the role of adjuvant chemotherapy (AC) in MIBC [58]. The results are still unclear because trials were inadequately underpowered, closed prematurely or unpublished. Table 4 summarizes the results of relevant clinical trials about AC in MIBC.

The EORTC 3099 trial evaluated AC (four cycles of CG, high-dose MVAC, or conventional MVAC) versus deferred six cycles cisplatin-based CT after surgery. A total of 284 patients with pT3-T4 or N+ M0 UC were randomized [59]. Although the primary endpoint of OS was not reached, AC showed significantly better DFS compared with deferred CT. However, this trial is controversial due to the design and the primary endpoint selected.

A total of 142 patients were included in a Spanish clinical trial that compared AC (paclitaxel, gemcitabine and cisplatin) with observation in resectable high-risk MIBC [60]. AC arm had a longer OS and DFS compared with observation arm, but this trial was prematurely finished because of the poor recruitment. An Italian clinical trial assessed the use of AC (CG) versus observation in patients with MIBC [61]. This trial also had poor recruitment, only 200 of the planned 600 patients were included and the difference in OS and DFS between the AC arm and the observation arm was not found.

Several systematic reviews and meta-analyses have been published, analyzing the data of adjuvant clinical trials in MIBC [58]. A meta-analysis of 10 clinical trials (1183 patients) showed a benefit of cisplatin-based adjuvant chemotherapy on OS (HR = 0.82, 95 % CI = 0.70–0.96, *p* = 0.02) [62]. An absolute improvement in survival from 50% to 56% was also observed at 5 years. In addition, cisplatin-based chemotherapy improved DFS (HR = 0.71, 95% CI = 0.60–0.83, *p* < 0.001), locoregional recurrence-free survival (HR = 0.68, 95% CI = 0. 55–0.85, *p* < 0.001) and metastasis-free survival (HR = 0.79, 95% CI = 0.65–0.95, *p* = 0.01).

Theoretically AC offers some advantages over NAC: early definitive treatment, such as RC; more accurate pathological staging and prognostic factor; and avoidance of overtreatment of some patients. However, AC has some limitations: patients treated with radical surgery are more probable to be ineligible for cisplatin-based AC because of impaired renal function or performance status [63].

### 3.2. Immunotherapy

Three randomized phase 3 clinical trials have evaluated the use of adjuvant immunotherapy (IT) in MIBC: IMvigor010, CheckMate 274 and AMBASSADOR [64,65].

The Imvigor010 trial compared atezolizumab with observation as adjuvant treatment in patients with high-risk MIBC in an open-label study [66]. A total of 807 patients were included with ypT2-4a or ypN+ tumors following NAC or pT3-4a or pN+ tumors if NAC was not administrated. Patients who had not been treated with NAC should decline or were ineligible for cisplatin-based adjuvant therapy. A total of 1200 mg of Atezolizumab was administered every three weeks for up to one year. DFS was the primary endpoint of this trial. No significant differences were found in DFS, 19.4 months in the atezolizumab arm vs. 16.6 months in the observation arm (HR: 0.89, 95% CI 0.74–1.08; *p* = 0.24).

Nivolumab as adjuvant treatment in patients with high-risk MIBC was evaluated in CheckMate 274 trial, a randomized, double-blind phase 3 clinical trial [67]. A total of 709 patients were included with ypT2-4a or ypN+ after NAC or pT3-4a or pN+ tumors if NAC was not administrated. Patients who had not been treated with NAC should decline or were ineligible for cisplatin-based AC. Inclusion criteria were similar in Imvigor010 and CheckMate 274. However, in the CheckMate trial, patients were randomized to receive 240 mg nivolumab every two weeks up to one year or a placebo. In this trial, the co-primary endpoints were DFS in all randomized patients and in patients with tumor PD-L1 expression ≥1%. In the whole population analysis, the DFS was 20.8 months in the nivolumab arm and 10.8 months in the control arm. The percentage of patients alive and disease free at 6 months was 74.9% in nivolumab group vs. 60.9 % in placebo group (HR: 0.70; 98.22 % CI, 0.55–0.90). Among patients with a PD-L1 expression ≥1%, the DFS at 6 months was 74.5% and 55.7% in the nivolumab and placebo groups, respectively (HR, 0.55; 98.72% CI, 0.35 to 0.85; *p* < 0.001).

AMBASSADOR is a phase 3 randomized clinical trial that evaluates pembrolizumab as adjuvant treatment in patients with high-risk MIBC (NCT03244384) [68]. The results of this study are not yet reported.

The results of the Imvigor010 and CheckMate 274 trials are summarized in Table 5. These two clinical trials are designed for the same population; however, they have some differences. Whereas the control arm in CheckMate 274 was the placebo group, in Imvigor010 it was the observation group. CheckMate 274 included more upper-tract UC patients than Imvigor010. Although the DFS of the ICI arm was similar in both trials (19.4 months with atezolizumab and 20.8 months with nivolumab), the DFS of their control groups was different (16.6 months in observation group in Imvigor 010 and 10.8 months in placebo group CheckMate 274). PD-1 blockade could be superior to PD-L1 blockade or other unknown factors may have varied significantly in the patient populations between these clinical trials. These results may be interpreted cautiously because it is not correct to compare two trials directly. For these reasons, the results of the AMBASSADOR clinical trial are expected. Moreover, adjuvant ICI with other agents is being investigated in the neoadjuvant trials [69].

## 4. Biomarkers

Several predictive and prognostic biomarkers have been evaluated in UC. However, there are not any useful biomarkers to guide the management of patients in the perioperative setting [69,70]. Many ongoing trials are currently exploring potential biomarkers; these results may improve the treatment of localized MIBC.

### 4.1. Biomarkers in NAC and AC Treatment

DNA damage response (DDR) gene alteration has been observed as a predictive factor of benefit to cisplatin-based NAC; a higher pathologic ORR and OS has been described in patients with this alteration [71,72,73,74,75]. Another study realized a whole exome sequencing analysis from 50 patients with MIBC treated with NAC. Among the 3227 genes detected with a potential somatic alteration, the only gene significantly associated with response in NAC group was ERCC2 [76]. Previously, ERCC2 mutations were associated with benefit to NAC [77]. However, it is unclear which DDR gene combinations provide the best predictive response [78]. In this way, there was not any association between response and alterations in ATM, RB1 or FANCC alone or in combination in a retrospective study of 165 patients treated with NAC and surgery [79].

Several prospective NAC trials (NCT02710734, NCT03558087 and Alliance A0311701) are evaluating DDR gene alterations (ATM, RB1, FANCC, and ERCC2) or more extensive DDR panels (ERCC2,3/BRCA1,2/RAD51C/ATR/ RECQL4/ATM/FANCC) as predictive factors in neoadjuvant setting [70]. For example, some trials are evaluating if patients with this alteration could conserve their bladders after neoadjuvant therapy if residual disease on clinical restaging are not demonstrated.

The role of p53 alterations is controversial. Alterations of p53 are associated with resistance to NAC [46]. It was associated with poor prognosis after RC and an elevated likelihood of response to AC. However, a phase 3 clinical trial randomized patients with p53-altered pT1-2N0M0 bladder carcinomas to adjuvant MVAC versus observation after RC. No differences based on p53 status were detected [80].

Circulating tumor DNA (ctDNA) is a known prognostic biomarker. The presence of ctDNA before starting and/or finishing NAC before RC is associated with prognosis. Moreover, the detection of ctDNA before RC was associated with residual disease after surgery. A total of 85% of patients that were ctDNA negative and 53% of patients who achieved negative ctDNA had pCR. No patients with positive ctDNA after NAC achieved pCR. In addition, the presence of ctDNA after surgery was associated with a higher probability of recurrence [81].

Several classifications of UC according to molecular alterations have been proposed in recent years. The actual consensus describes six subtypes of UC based in previous classifications: luminal papillary, luminal not-specified, luminal unstable, stroma-rich, basal/squamous and neuroendocrine-like [82]. Basal/squamous tumors are characterized by the presence of markers of basal cells (cytokeratin 5, 6, 14), mutations in RB1 and TP53, squamous features and greater tumor aggressiveness with a predominance of more advanced stages. Luminal tumors express urothelial differentiation biomarkers (GATA3, FOXA1), with alterations in FGFR3. In this type of tumor, the diagnosis is made in earlier stages with better prognosis and predominantly in young patients. Several studies have analyzed the association between the different tumor subtypes and the response to NAC. Choi et al. defined a third subtype called p53-like associated with resistance to CT [46]. In the phase 2 study in patients with MIBC treated with NAC (M-VAC and bevacizumab), a better prognosis was detected in patients with basal tumors with respect to the luminal and p53-like subtypes, respectively [83].

Seiler et al. proposed a specific molecular classification for each tumor based on four subtypes (basal, luminal, luminal-infiltrate, and claudin-low) [84]. The luminal patients had the best prognosis, in contrast to the claudin-low subtype that had the worst prognosis regardless of the use of NAC. A significant improvement in survival was observed in patients with basal tumors treated with NAC compared to those treated with surgery alone. A study evidenced the benefit of NAC in patients with non-luminal subtypes (basal, claudin-low and luminal-infiltrate), while in luminal tumors, the administration of NAC did not significantly change the prognosis [85]. A study in which molecular classification was established using immunohistochemical staining (CK 5/6, CK14, CK20, GATA3, FOXA1), patients classified as baseline were more likely to present a pCR than patients with luminal tumors [86].

Finally, several analytic values in peripheral blood may be prognostic biomarkers in patients treated with CT in MIBC: neutrophil/lymphocyte ratio (NLR), platelet/lymphocyte ratio (PLR), hemoglobin and albumin [87,88,89,90].

In summary, there are no conclusive data to select patients who may benefit from NAC and AC treatment based on their molecular classification or another biomarker.

### 4.2. Biomarkers in IT Treatment

PD-L1 expression, tumor mutation burden (TMB) and ctDNA are potential biomarkers for IT in perioperative treatment of MIUC [91].

PD-L1 expression is a controversial predictive biomarker in UC because diverse results have been reported in mUC, according to PD-L1 expression [92,93,94,95,96,97]. These heterogenous results may be explained by different methods for scoring PD-L1, cut off values of PD-L1 expression and changes in PD-L1 expression in UC. Moreover, higher rates of tumor mutation burden (TMB) are associated with better responses to ICI in mUC [91,92].

An analysis of biomarkers has been realized in the clinical trials of neoadjuvant IT in perioperative treatment of MIBC. PD-L1 expression and the elevated TMB (>15 mut/Mb) of mutations in DDR gene alteration were associated with higher ORR in PURE-01 [22]. A higher pCR was detected in tumors with expression of PD-L1 and high TMB in the NABUCCO trial [25]. However, these biomarkers were not associated with response to atezolizumab or durvalumab and tremelimumab [29,98]. DUTRENEO trial did not detect any association between pCR and PD-L1 expression or TMB [28]. Therefore, the analysis of biomarkers reported in these clinical trials do not determine which patients may benefit from neoadjuvant IT in the perioperative treatment of MIBC.

The ctDNA was determined before adjuvant atezolizumab and 6 weeks after the start of adjuvant treatment in the iMvigor010 trial [99,100]. A total of 37 % of patients had ctDNA positivity (ctDNA+). The ctDNA+ patients showed better DFS and OS with atezolizumab vs. observation. In those positive patients, a survival benefit was also detected in PD-L1-high, and TMB-high patients (HR = 0.52, 95% CI 0.331–0.82, *p* = 0.004; HR = 0.34, 95% CI 0.19–0.6, *p* < 0.0001, respectively). Furthermore, post-surgical ctDNA+ would be useful to identify minimal residual disease and, in consequence, patients who would probably benefit from adjuvant atezolizumab. These results support IMvigor011 trial design, which will assess atezolizumab as adjuvant therapy in ctDNA+ patients (NCT04660344).

Although it is not a biomarker, sex differences have been observed on the tumor immune microenvironment, especially in NMIBC [101,102]. These differences could have an impact on the benefits observed with new immunotherapies. However, other cofounding factors, such as tobacco or clinical management, could influence this consideration.

Currently, there is not any useful or established biomarkers for identifying patients that will benefit from immunotherapy in the perioperative setting of MIBC.

### 4.3. Urinary Biomarkers

Significant progress in urinary biomarkers for BC have been developed, especially in NMIBC. Urinary biomarkers could be useful in combination with cystoscopy and imaging techniques in the follow-up of the patients in the perioperative setting. Although FDA has approved several urinary tests in NMIBC (NMP22, NMP22 BladderCheck, UroVision, BTA-TRAK, BTA-STAT, ImmunoCyt), the international guidelines do not recommend their clinical use [103]. The protein expressions of novel urinary biomarkers, which target genetic and epigenetic alterations, are being studied in NMIBC [104].

Furthermore, urinary biomarkers in MIBC after neoadjuvant treatment have the additional challenge of differentiating post-treatment changes in bladder urothelial cells. There are several ongoing trials that explore potential urinary biomarkers in MIBC; in this way, these analyses could help us to evaluate the response of our patients early during the treatment [105].

The genitourinary microbiome could be related with UC pathogenesis and progression and act as a non-invasive and modifiable urinary biomarker [106,107]. The genitourinary and gastrointestinal microbiome could be a prognostic and predictive biomarker in patients treated with ICI as perioperative treatment for MIBC, but its role is still unknown [108]. For example, the use of antibiotics is associated with a lower percentage of CR in patients treated with pembrolizumab as neoadjuvant treatment [109]. New studies are necessary for defining the relationship between the urinary microbiota and the effectiveness of ICI in perioperative settings.

## 5. RC-Avoidance

Although RC is the current standard of care, it is a complex procedure, associated with significant postoperative complications, morbidity, mortality, alteration of body image and deterioration of quality of life, even with significant advances in minimally invasive surgery [110,111]. Patients may also have contraindications to this procedure, so bladder-sparing strategies are needed. Different approaches have been evaluated and perioperative systemic treatment may play a role in these patients. It is important to note that a significant proportion of patients will achieve a pCR after NAC, as described before. These patients may spare the surgical procedure, but close monitoring is needed. This approach has been tested in two studies. In a retrospective study, 143 patients with T2-4 N0 BC were treated with three cycles of MVAC. A total of 49% of patients achieved a cT0 response. After re-staging, 52 patients underwent TURB alone, 13 patients underwent partial cystectomy and 39 patients underwent RC. A total of 60% of patients who underwent TURB alone were alive at a median follow-up of 56 months and 44% in that TURB group did not need salvage cystectomy [112]. Another retrospective study of patients achieving cT0 showed a five-year cancer-specific survival of 87%, 5-year DFS of 58% and 5-year cystectomy-free survival of 79% [113]. However, there is currently no standard biomarker that may help to identify good responder patients; therefore, a pre- and post-treatment identification is lacking.

Prospective randomized studies have been developed to answer this question and to find out the role of genetic mutations in the bladder-sparing strategies in MIBC. RETAIN, RETAIN-2 and Alliance phase II clinical trial A031701 will evaluate risk-adapted approaches based on response to neoadjuvant chemotherapy (RETAIN and A031701) or chemoimmunotherapy (RETAIN-2). An interim analysis of RETAIN trial has been published. A total of 76% of patients with a mutation in ATM, RB1, ERCC2 and FANCC were cT0 at post-NAC TURBT, with a bladder preservation rate of 89%. With a median follow-up of 14.9 months, 50% of patients with mutations in these genes that followed active surveillance have recurred. Of these relapses, 50% were NMIBC and 50% were MIBC, locally advanced or metastatic [114].

## 6. Conclusions and Future Perspective

Neoadjuvant cisplatin-based chemotherapy followed by RC is the current standard of care for local MIBC. However, the risk of recurrence is still high, and many patients are not candidates for this strategy.

The development in UC in recent years has been higher than in the three previous decades. The appearance of new therapies in mUC provides new opportunities of treatment. Currently, there are several phase 3 clinical trials that are evaluating potential new standards of care in localized and advanced urothelial carcinoma. New systemic combinations that include immunotherapies with other more sophisticated drugs with new mechanism of actions, such as ADCs or targeted therapy, can potentially replace cisplatin-based chemotherapy as the standard therapy in urothelial carcinoma. These results can modify treatment algorithms in UC from NMIBC to metastatic disease.

The data of phase 2 trials with these drugs in the perioperative setting are promising. Currently, phase 3 studies are evaluating their role as neoadjuvant and adjuvant treatment. These new agents can change the landscape of MIBC, improving the outcomes of standard treatments and offering therapeutic alternatives in unfit patients.

## Figures and Tables

**Table 3 cancers-15-00566-t003:** Summary of completed, active, or currently recruiting studies with immunotherapy agents in combination with chemotherapy.

Study	Phase	Treatment	Patients Included	pCR	Survival
HCRN GU 114-88 (Cohort 1)	1b/2	CG + PEM	43	44.4%	Estimated 36 m RFS: 63% Estimated 36 m OS: 82%
HCRN GU 114-88 (Cohort 2)	1b/2	G + PEM	37	45.2%	Estimated 12 m RFS: 74.9% Estimated 12 m OS: 93.8%
lccc1520 trial	2	C (35)G + PEM	39	36%	-
BLASST-1	2	CG + N	41	65.8 (≤ypT1N0)	12 m PFS: 85.4%
SAKK 06/17	2	CG + DUR	61	34%	OS at 2 years: 87.3%
NCT02989584	2	AZ → CG + AZ → AZ	44	41%	No relapse in <ypT2N0 patients.
AURA (cohort 1)	2	CG + Av vs. dd-MVAC +Av	28 28	32% 43%	- -
NIAGARA	3	CG + DUR → DUR	1050 to include	-	-
KEYNOTE 866	3	PG + PEM → PEM	870 to include	-	-
ENERGIZE	3	PG + N ± LM	1200 to include	-	-
SWOG-GAP	2	Ca + G + Av	196 to include	-	-
NEMIO	2	ddMVAC + DUR	120 to include	-	-
RETAIN-2	2	AMVAC + N	71 to include	-	-
NCT04383743	2	MVAC + PEM	17 to include	-	-

Abbreviations: Av: avelumab, AMVAC: accelerated MVAC, C: cisplatin, Ca carboplatin, ddMVAC: dose-dense MVAC, DUR: durvalumab, G: gemcitabine, LM: linrodostat mesylate, N: nivolumab, OS: overall survival, pCR: pathological complete response PEM: pembrolizumab, RFS: recurrence-free survival.

**Table 4 cancers-15-00566-t004:** Summary of phase 3 trials for adjuvant chemotherapy in MIBC.

	EORTC 30994 (NCT00028756) [59]		SOGUG 99/01 [60]		Italian Multicenter Trial [61]	
Treatment	CG, high-dose MVAC, MVAC	Deferred CT	PGC	Observation	CG	Observation
*n*	141	143	68	74	102	92
Median follow-up	7 years		30 months		35 months	
5-year OS (%)	53.6	47.7	60	31	43.4	53.7
mOS (y)	6.74	4.60	NR	NR	NR	NR
5-year PFS (%)	47.6	31.8	NR	NR	37.2	42.3
Median PFS (y)	3.11	0.99	NR	NR	NR	NR
Percent progressed (%)	45	62	NR	NR	NR	NR
Limitations	Underpowered to detect OS and DFS benefit with slow accrual and premature termination. The pT2 patients were not enrolled. No central pathology review. Patient reported outcomes were not recorded.		Small sample size Poor accrual Premature closure		Small sample size Poor accrual Underpowered Premature closure Poor compliance with AC No central pathologic review	

Abbreviations: CG: cisplatin and gemcitabine, MVAC: Methotrexate, Vinblastine, Doxorubicin, and Cisplatin, PGC paclitaxel, gemcitabine, and cisplatin NR: not reported, OS: overall survival, PFS: progression-free survival, y: years.

**Table 5 cancers-15-00566-t005:** Summary of phase 3 trials for adjuvant immunotherapy in MIBC.

	Imvigor010 (NCT02450331) [66]	CheckMate 274 (NCT02632409) [67]
Agent	Atezolizumab	Nivolumab
Control	Observation	Placebo
*n*	809	709
% upper-tract UC	6.6	21
Primary endpoint	DFS	DFS
DFS about PD-L1	No	Yes
DFS	19.4 m vs. 16.6 m	20.8 m vs. 10.8 m
Primary endpoint obtained	No	Yes
Grade 3-4 TRAE	16%	17.9% vs. 7.2%

Abbreviations: m: month, MIBC: muscle-invasive bladder cancer, NAC: neoadjuvant chemotherapy, PD-L1: programmed death ligand 1, TRAEs: treatment-related adverse events, UC: urothelial cancer.

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
