# Peer review of "Current and Future Landscape of Perioperative Treatment for Muscle-Invasive Bladder Cancer"

_cancers, 2023, doi:10.3390/cancers15030566_

Round 1
Reviewer 1 Report
I congratulate the authors of this review neoadjuvant and adjuvant therapy for patients with MIBC. The paper is clear and comprehensively describes the possible treatment options, accurately reporting the results of recent trials. I particularly find the paragraph on biomarkers of great interest. However, It should be noted that the topic of the review is not new, and the authors provide no innovative insights.
Author Response
First of all I would like to thank your comments. Probably, in the next months we have new data of different trial in urothelial carcinoma. In this way, we thought that our review that resume the current evidence of perioperative treatment could be useful to put in context the potential changes in the management of urothelial carcinoma
Reviewer 2 Report
Well written and timely review on the MIBC landscape.
Author Response
I would like to thank your review; I appreciate so much your comment
Reviewer 3 Report
The authors have done great job in reviewing current perioperative treatment options for MIBC patients. The manuscript is written well with minor grammar errors.
Inclusion of sex as a biological variable in this discussion should make this review exceptional.
It would be nice if the authors could add the future perspective discussion section and talk about immunotherapy and role of microbiome in these patients.
Also adding a few sentences about molecular biomarkers in urine for informed management of bladder cancer, RC-avoidance, and tumor-stage determination would make this manuscript better.
Author Response
We have included a discussion about sex as a biological variable in IT biomarker paragraph
We have added a section of urinary biomarkers in which we also discuss the role of microbiome
Another section about RC-avoidance has been included
A future perspective discussion has been included in conclusions section